# Physicochemical Characteristics, Techno-Functionalities, and Amino Acid Profile of *Prionoplus reticularis* (Huhu) Larvae and Pupae Protein Extracts

**DOI:** 10.3390/foods12020417

**Published:** 2023-01-16

**Authors:** Ruchita Rao Kavle, Patrick James Nolan, Alaa El-Din Ahmed Bekhit, Alan Carne, James David Morton, Dominic Agyei

**Affiliations:** 1Department of Food Science, University of Otago, Dunedin 9054, New Zealand; 2Department of Biochemistry, University of Otago, Dunedin 9054, New Zealand; 3Department of Wine, Food and Molecular Biosciences, Lincoln University, Lincoln 7647, New Zealand

**Keywords:** amino acid profile, edible insect material, Huhu grub, physiochemical properties, *Prionoplus reticularis*, protein extracts, techno-functionalities

## Abstract

The amino acid profile, techno-functionalities (foaming stability/capacity, emulsion stability/capacity, solubility, and coagulation), and physicochemical characteristics (colour, particle size, surface hydrophobicity, Fourier-transform infrared spectroscopy, and differential scanning calorimetry) of protein extracts (PE) obtained from *Prionoplus reticularis* (Huhu grub) larvae (HLPE) and pupae (HPPE) were investigated. Total essential amino acid contents of 386.7 and 411.7 mg/g protein were observed in HLPE and HPPE, respectively. The essential amino acid index (EAAI) was 3.3 and 3.4 for HLPE and HPPE, respectively, demonstrating their nutritional equivalence. A unique nitrogen-to-protein conversion constant, k, and the corresponding protein content of the extracts were 6.1 and 6.4 and 72.1% and 76.5%, respectively. HLPE (37.1 J/g) had a lower enthalpy than HPPE (54.1 J/g). HPPE (1% *w/v*) exhibited a foaming capacity of 50.7%, which was higher than that of HLPE (41.7%) at 150 min. The foaming stability was 75.3% for HLPE and 73.1% for HPPE after 120 min. Both protein extracts (1% *w/v*) had emulsifying capacities that were 96.8% stable after 60 min. Therefore, protein extracts from Huhu larvae and pupae are of a good nutritional quality (based on their EAAI) and have techno-functional properties, such as foaming and emulsification, that afford them potential for certain food technology applications.

## 1. Introduction

Edible insects and their juvenile forms have recently been recognised as potential alternative foods that are rich in nutrients such as lipids, carbohydrates, and protein. The protein content of edible insects is comparable to that of conventional food sources such as dairy, meat, egg, and soy [1]. Aside from their nutritional benefits, proteins play a crucial role in defining food quality in relation to functionalities including foaming, gelling, emulsification, and the oil- and water-holding capacity [2]. The functionalities of protein extracts obtained from edible insects such as *Tenebrio molitor*, *Apis mellifera*, *Schistocerca gregaria*, *Hermetia illucens*, and *Rhynchophorus ferrugineus* have been reported [3,4,5,6]. These studies established that edible insect protein exhibits functionalities similar to those of protein from conventional sources and that these functionalities can be substantially affected by the techniques used to extract the protein [3].

Providing insect material as a protein extract in powder form, for example, can help to overcome consumer acceptance barriers [7], and it is reported that people are more willing to consume processed insect material rather than whole insects/grubs [3].

The study of protein functionalities can be facilitated by extracting protein from the natural product source material. Protein recovery has historically been estimated through indirect approaches such as the Kjeldahl method, where the nitrogen content is determined and then converted to crude protein using a nitrogen-to-protein conversion factor, such as the factor of 6.25 used for meat [8]. However, Janssen, et al. [9] proposed that the protein content determination of insect material should be calculated using species-dependent factors because of the presence of relatively high levels of non-protein nitrogen in insects. Consequently, the nitrogen-to-protein factors for the protein extracts of various insects, such as *T. molitor* larvae, *Acheta domesticus*, *Locusta migratoria*, *A. mellifera*, and *S. gregaria*, have been reported [8,9].

To date, the nitrogen-to-protein conversion factors and protein functionalities of indigenous insect species such as *Prionoplus reticularis* (commonly known as Huhu) have not been reported. Huhu grubs have a long history of use as food among the indigenous Māori community of New Zealand, and we recently reported, for the first time, the proximate composition and mineral profile of four developmental stages of *P. reticularis* [10,11].

Huhu grubs are usually consumed as large larvae. However, a comparative assessment of the properties at various developmental stages is central to establishing dietary and nutritional guidelines for Huhu grubs as an alternative food protein source and their prospective use in food technology applications. This research, therefore, aimed to study the physicochemical characteristics (colour, particle size, surface hydrophobicity, FTIR, and DSC), techno-functionalities (foaming stability/capacity, emulsion stability/capacity, solubility, and coagulation), amino acid profiles, and essential amino acid indices of protein extracts from Huhu larvae and pupae.

## 2. Materials and Methods

### 2.1. Sample Collection

*Prionoplus reticularis* large larvae and pupae (Figure 1) were collected from dead *Pinus radiata* (Pine) logs at Flagstaff, Three Mile Hill, which is a locality near Dunedin in the Otago Region (South Island) of New Zealand (latitude 45.8656 and longitude 170.3785). Huhu grubs can be identified by their physical characteristics, including the terminal triangular spine and setae on the abdominal segments in the large larval stage. Huhu pupae can be identified by their flexible abdominal segments and substantially darker bodies after being removed from decaying wood and exposed to light and the open-air environment (Edwards, 1961a). The Te Ara Encyclopedia of New Zealand has pictures and descriptions of Huhu larvae and pupae (see https://teara.govt.nz/en/photograph/14355/huhu-grub, accessed on 14 August 2022). As reported in our previous study [11], Huhu larvae (average length 3.6 cm, weight 4.3 g) and pupae (average length 2.4 cm, weight 3.6 g) harvested from the same geographical area were found to be composed of 58.7% and 66.6% moisture (wet weight), 1.8% and 2.2% ash (dry weight, DW), 45% and 58.4% DW fat, and 30.1% and 27.6% DW protein, respectively.

### 2.2. Huhu Grub Protein Extraction Yield

After freeze-drying (LABCONCO, FreeZone 12 Plus, KCMO, Kansas City, MO, USA, collection temperature −84 °C, vacuum pressure 200 Pa) for 48 h, the Huhu grub larvae and pupae protein was extracted by alkaline extraction and isoelectric pH precipitation, according to a method reported by Mishyna, Martinez, Chen and Benjamin [8], with slight modification. Aliquots of raw powder (200 g) were dispersed in a chloroform: methanol (2:1) solution by stirring for 1 h on a magnetic stirrer (Lab supply, Benchmark Scientific, Dunedin, New Zealand). The chloroform fraction containing fat was removed after the phase separation, and the powder was re-extracted using the same technique until clear phases were obtained. The defatted insect powder was air-dried at room temperature overnight and stored in containers at 20 °C prior to use. The defatted and air-dried Huhu grub material was ground and sieved using a mesh size of 425 µm. Aliquots of the ground material were dispersed 1:25 (*w/v*) in 0.25 M NaOH at 40 °C, stirred at 200 rpm for 20 min, and then centrifuged at 3370× *g* using a Beckman-Coulter Allegra X-15R centrifuge (USA) for 10 min at 4 °C. The supernatant (~pH 10.8) was adjusted to pH 5 (the pI of the protein extract was previously determined experimentally) and then centrifuged (3000× *g*, 4 °C, 10 min). The pellet was retained and washed twice with deionised water (pH 5) and then freeze-dried (LABCONCO, FreeZone 12 Plus, USA). The yield of the extracted protein obtained from the insect material was calculated using Equation (1).
(1)$$\mathrm{Protein}\;\mathrm{extract}\;\mathrm{yield}\;\left(\%\right)=\frac{\mathrm{Weight}\;\mathrm{of}\;\mathrm{the}\;\mathrm{dried}\;\mathrm{protein}\;\mathrm{powder}}{\mathrm{Total}\;\mathrm{weight}\;\mathrm{of}\;\mathrm{the}\;\mathrm{dried}\;\mathrm{insect}\;\mathrm{material}\;}\times 100$$

The nitrogen content of the protein extract was determined by the Kjeldahl method, and the crude protein content was calculated using a new nitrogen-to-protein conversion factor calculated for Huhu grub protein extract (see Section 2.4). The Huhu larvae and pupae protein extracts are referred to as HLPE and HPPE in the subsequent sections.

### 2.3. Amino Acid Profiles of Huhu Grub Protein Extracts

The amino acid profiles of HLPE and HPPE were determined using the method of Jayawardena, et al. [12]. The freeze-dried HLPE and HPPE were ground, sieved to a 0.48 mm mesh size, and stored in airtight containers. Aliquots of the HLPE and HPPE powder (3 mg) were resuspended in 5 mL of 6 M HCl, and 10 μL of 0.5 M aminobutyric acid was added as an internal standard. The samples were mixed by vortexing and then sonicated (5 min, room temperature), and each tube was purged with oxygen-free nitrogen, immediately airtight-sealed, and then hydrolysed at 110 °C for 20 h. The hydrolysates were cooled and dried in a rotary vacuum evaporator at 45 °C, and the residue was resuspended in Type 1 MilliQ water and then transferred to a 50 mL volumetric flask. The aliquots of the samples were filtered (0.45 μm) prior to amino acid analysis using an Agilent 1100 series HPLC (Agilent Technologies, Waldbronn, Germany) equipped with an Agilent autosampler and fluorescence detector and an ACE 3 μm C-18 (150 mm × 4.6 mm) column at 40 °C, according to the method of Jayawardena, Morton, Brennan and Bekhit [12]. The amino acid standard was obtained from Sigma-Aldrich (Sigma-Aldrich, Darmstadt, Germany).

### 2.4. Determination of the Essential Amino Acid Index (EAAI) and Nitrogen-Protein Conversion Factor

The protein quality was evaluated using the essential amino acid index (EAAI), which is based on the content of all the essential amino acids compared to a reference protein. The EAAI provides a measure of the quality of a food protein for human consumption and was calculated using Equation (2), as reported by Yang, et al. [13]
(2)$$\mathrm{EAAI}=\sqrt[{n}]{\frac{\mathrm{mg}\;\mathrm{of}\;\mathrm{lysine}\;\mathrm{in}\;1\;g\;\mathrm{of}\;\mathrm{Huhu}\;\mathrm{protein}}{\mathrm{mg}\;\mathrm{of}\;\mathrm{lysine}\;\mathrm{in}\;1\;g\;\mathrm{WHO}\;\mathrm{daily}\;\mathrm{human}\;\mathrm{adult}\;\mathrm{requirement}}\times \left(\ldots \mathrm{other}\;\mathrm{essential}\;\mathrm{amino}\;\mathrm{acid}\right)}$$

For our study, the mg/g daily adult human requirement for essential amino acids as recommended by the FAO/WHO [14] was used as the reference protein (Table 1).

### 2.5. Determination of the True Protein and Nitrogen-to-Protein Conversion Factors for Huhu Protein Extracts

The amino acid analysis data were converted to the mass concentration using the respective ‘in-chain’ molecular weight of each amino acid. The ‘true protein’ contents (in g/100 g) of HLPE and HPPE were expressed as the sum of the amino acid content [18]. The nitrogen-to-protein conversion factor, *k*, was calculated as reported by Mishyna, Martinez, Chen and Benjamin [8], as the average of *k*_A_ and *k*_P_. *k*_A_ was calculated as the ratio of the sum of the amino acid residues to the weight of total protein nitrogen, and *k*_P_ was calculated as a ratio of the sum of amino acid residues to the total nitrogen content, as determined by the Kjeldahl method [18].

### 2.6. Colour Measurement of Huhu Grub Protein Extracts

The colour of the *P. reticularis* dried protein extracts was measured in triplicate using a Hunterlab miniscan XE Spectrocolorimeter (Hunterlab, Reston, VA, USA) that recorded *L** (lightness), *a** (red/green), and *b** (blue/yellow). Before measuring the samples, the colorimeter was calibrated with white and black standard tiles.

Following this, Equations (3) and (4) were used to calculate the browning index (B.I.).
(3)BI=[100 × (x − 0.31)]/0.17
(4)x = (a* + 1.75 × L*)/(5.645 × L* + a* − 3.012 × b*)

### 2.7. Particle Size Analysis of the Huhu Grub Protein Extract Suspensions

A suspension (1% *w/v*) of HLPE and HPPE powder was prepared in 5 mL of 2 mM sodium phosphate (pH 7) and then incubated for 1 h at 25 °C, followed by syringe filtering through a 0.45 m pore size (Merck, Darmstadt, Germany). Measurements were performed at a scattering angle of 173° and a temperature of 25 °C. The particle size distribution in the protein suspension was measured in triplicate using a Nano Series Zetasizer (Malvern, Australia), according to Queiroz, et al. [19].

### 2.8. Surface Hydrophobicity of the Huhu Grub Protein Extract Suspensions

The protein surface hydrophobicity was measured according to Mishyna, Martinez, Chen and Benjamin [8], with slight modification. A suspension (1% *w/v*) of HLPE and HPPE powder was prepared in 5 mL of 2 mM sodium phosphate (pH 7) and then incubated for 1 h at 25 °C. The suspension was centrifuged (3000× *g*, 20 min, 20 °C), and the protein concentration of the supernatant was estimated using the Lowry assay [20]. To 2 mL aliquots of samples diluted to 0.0025, 0.005, and 0.01% (*w/v*) we added 10 µL of 8-anilinonaphthalene-1-sulfonic acid (ANS) solution (Sigma-Aldrich, Darmstadt, Germany) and then vortexed and incubated the samples at room temperature in the dark for 15 min. The fluorescence intensity was measured using a Synergy 2 Microplate Reader (USA) (excitation 390 nm, emission 470 nm). The surface hydrophobicity value was derived from the slope of a linear regression plot of the protein concentration and fluorescence intensity.

### 2.9. Differential Scanning Calorimetry (DSC) of the Huhu Grub Protein Extracts

Differential scanning calorimetry (DSC) analysis was conducted to measure the thermal stability of the defatted HLPE and HPPE according to Queiroz, Regnard, Jessen, Mohammadifar, Sloth, Petersen, Ajalloueian, Brouzes, Fraihi and Fallquist [19]. Triplicate measurements were performed using a DSC 250 instrument (T.A. Instruments Ltd., New Castle, DE, USA) with 4 mg aliquots of protein extract powder. The samples were scanned between 10 °C and 250 °C, maintaining a heating rate of 5 °C/min. The onset (T_o_), peak (T_p_), and conclusion temperatures (T_c_), change in enthalpy (ΔH, calculated by integrating the area under the endothermic peak), and temperature range (ΔT_d_ = T_c_ − T_o_) associated with the denaturation of both protein extracts were calculated using the T.A. Universal Analysis 200 software (T.A. Instruments Ltd., New Castle, DE, USA).

### 2.10. Fourier-Transform Infrared Spectroscopy (FTIR) of the Huhu Grub Protein Extracts

To obtain a biochemical fingerprint of the HLPE and HPPE, FTIR spectra of the defatted protein extracts were obtained by the method of Queiroz, Regnard, Jessen, Mohammadifar, Sloth, Petersen, Ajalloueian, Brouzes, Fraihi and Fallquist [19] using a Bruker Optics Fourier-transform infrared (FTIR) spectrometer (Alpha Systems, Waltham, MA, USA) fitted with an attenuated total reflection (ATR) platinum diamond 1 accessory. Triplicate measurements of 5 mg aliquots of protein extract powder were obtained over 4 scans within the transmission range of 4000–650 cm^−^^1^, and the spectra were plotted as a function of the wavenumber (cm^−^^1^).

### 2.11. Foaming Capacity and Stability of the Huhu Grub Protein Extract Suspension

The foaming capacity and foam stability of the HLPE and HPPE suspensions were determined according to Mishyna, Martinez, Chen and Benjamin [8], with slight modifications. A 10 mL 1% (*w/v*) suspension of Huhu grub protein extract powder in 0.2 M sodium phosphate (pH 7) was mixed in an orbital shaking incubator (Ratek, Boronia, Australia) at 25 °C for 1 h and then homogenised for 2 min at 12,000 rpm using a T25 Digital Ultra Turrax (IKA, Melbourne, Australia). The foaming capacity was measured by recording the volume of the foam layer 10 s after homogenisation. The foaming capacity was calculated using Equation (5), where H_0_ is the initial sample volume (mL) and H_t_ is the sample volume following homogenisation. The foam stability was determined by recording the volume of the foam layer at 5, 10, 15, 30, 60, 90, and 120 min after homogenisation. The foam stability was then calculated using Equation (6), where F.C. is the foaming capacity and F.C._0_ is the foaming capacity at time zero.
(5)$$\mathrm{Foaming}\;\mathrm{capacity},\;\%=\frac{H_{t}-H_{0}}{H_{0}}\times 100$$
(6)$$\mathrm{Foaming}\;\mathrm{stability},\;\%=\frac{F.C.}{F.C._{0}}\times 100$$

### 2.12. Water- and Oil-Holding Capacity of the Huhu Grub Protein Extract Suspensions

The water- and oil-holding capacity (WHC, OHC) was measured following the method of Clarkson, Mirosa and Birch [7]. Aliquots (0.25 g) of the Huhu grub protein extract powder were either added to 5 mL of Type 1 MilliQ water (for the determination of WHC) or 5 mL of 100% pure sunflower oil (Pams, Auckland, New Zealand) for the determination of OHC and centrifuged (2200× *g*, 10 min, 4 °C) (Beckman, Coulter Ltd., Pao Alto, CA USA). The supernatant was carefully decanted and measured. The water-/oil-holding capacity, expressed as mL/g, was calculated according to Equation (7).
(7)$$\mathrm{Water}-\;\mathrm{or}\;\mathrm{oil}-\mathrm{holding}\;\mathrm{capacity}\;\left(\frac{\mathrm{mL}}{g}\right)=\frac{\mathrm{water}\;\mathrm{or}\;\mathrm{oil}\;\mathrm{held}\;\left(\mathrm{mL}\right)}{\mathrm{Sample}\;\mathrm{weight}\;\left(g\right)}\;$$

### 2.13. Emulsifying Activity/Capacity of the Huhu Grub Protein Extract Suspensions

The emulsifying capacity (EC) and emulsion stability (ES) of the HLPE and HPPE suspensions were determined according to Kim, et al. [21], with slight modification. A 1% (*w/v*) suspension was prepared by combining 0.1 g of HLPE and HPPE with 10 mL of 0.2 M sodium phosphate (pH 7), followed by mixing with 1 mL of 100% pure sunflower oil (Pams, Auckland, New Zealand) in a 50 mL Falcon tube and homogenisation (T25 Digital Ultra Turrax, IKA, Melbourne, Australia) at 18,000 rpm for 2 min. The volume of the emulsified oil layer was measured 10 min after homogenisation, and the emulsifying capacity was measured according to Equation (8), where H_f_ is the initial sample volume (mL) and H_i_ is the sample volume following homogenisation.

The emulsion stability was determined by immediately dispersing 50 µL of the emulsion after homogenisation in 10 mL of 0.3% (*w/v*) SDS solution. Following gentle inversion, the light absorbance of the mixture at 500 nm was measured at 10, 20, 30, 60, 90, and 120 min after incubation. The emulsion stability was calculated according to Equation (9), where E_0_ is the absorbance of the emulsion at 500 nm at time zero, and E_t_ is the absorbance at various time intervals.
(8)$$E\mathrm{mulsifying}\;\mathrm{capacity}\;\left(\mathrm{EC}\right),\;\%=\frac{H_{f}}{H_{i}}\times 100$$
(9)$$\mathrm{Emulsion}\;\mathrm{stability}\;\left(\mathrm{ES}\right),\;\%=\frac{E_{0}-E_{t}}{E_{0}}\times 100$$

### 2.14. Solubility Profile of the Huhu Grub Protein Extract Suspensions

The solubility was measured using the method of Mishyna, Martinez, Chen and Benjamin [8], with slight modification. A 1% (*w/v*) suspension of HLPE and HPPE powder was prepared in 5 mL of Type 1 MilliQ water. Aliquots of the suspension were adjusted to a pH between 3 and 9 and then incubated for 1 h, followed by centrifugation for 20 min at 3000× *g*, and the supernatants were collected for the measurement of protein concentration, determined by the Lowry assay [20]. The protein solubility was calculated as the ratio of soluble protein in the supernatant compared to the total crude protein in the HLPE and HPPE samples.

### 2.15. Coagulation of the Huhu Grub Protein Extracts

The heat-induced coagulation of the HLPE and HPPE suspensions was performed as reported by Mishyna, Martinez, Chen and Benjamin [8], with slight modification. A 1% (*w/v*) suspension of HLPE and HPPE powder was prepared in 10 mL of sodium citrate phosphate (pH 7), shaken for 5 min in an orbital shaking incubator (Ratek, Boronia, Australia), and then centrifuged for 15 min at 3500× *g* at room temperature. To prepare the ‘before heating’ controls, 8 mL of Biuret reagent was added to 2 mL of the supernatant and then incubated in the dark for 30 min. The absorbance at 540 nm was recorded using a Synergy 2 Microplate Reader (USA). The ‘after heating’ samples were prepared by heating the remaining supernatant for 15 min in a water bath (Polysceince, Auckland, New Zealand) at 100 °C, followed by cooling to room temperature. The protein coagulation (%) was calculated using Equation (10), where *A*_0_ is the absorbance at 540 nm before heating, and A_.H._ is the absorbance after heating.
(10)$$\mathrm{Coagulation},\;\%=\frac{A_{0}-A_{t}}{A_{0}}\times 100$$

### 2.16. Statistical Analysis

All the experiments were conducted using three independent replicates. The results were then subjected to analysis of variance, performed using Minitab^®^ version 16.1 (Minitab Limited, Sydney, Australia), and significant differences were confirmed at *p* < 0.05.

## 3. Results and Discussion

### 3.1. Huhu Grub Protein Extract Yield and Protein Recovery

The yields of the alkaline protein extraction of freeze-dried whole Huhu larvae and pupae were 31.9% DW and 33.5% DW, respectively (Appendix A). The higher yield of protein obtained from the pupae was statistically significant (*p* = 0.02). These results obtained for Huhu grubs are higher than those reported for alkaline-extracted proteins from *A. mellifera* (larvae) and *Schistocerca gregaria* (adult), which yielded 27.5% DW and 24.2% DW, respectively [8], and *Rhynchophorus ferrugineus*, which yielded 65.7% [6]. Moreover, the protein recovery for the Huhu larvae and pupae was 72.1% and 76.5% DW using nitrogen-to-protein conversion factors of 6.1 and 6.4, respectively, which were determined in this study (see Section 3.4). The protein portion of the material extracted from the Huhu larvae and pupae powder was found to be greater than 65% on the basis of the dry weight; hence, the extracted material can be referred to as a ‘protein extract’.

### 3.2. Amino Acid Composition

The amino acid profiles of the Huhu larvae protein extract (HLPE) and Huhu pupae protein extract (HPPE) are shown in Table 1. Nineteen amino acids, including eight essential amino acids, were identified in the HLPE and HPPE (Table 1). The total essential amino acid (TEAA) content was 386.7 mg/g protein for HLPE and 411.7 mg/g protein for HPPE. The Huhu larvae and pupae protein extract TEAA values were higher than those reported for other insect species, such as wasp (*Brachygastra mellifica*) larvae (305.5 mg/g protein) [22] and mealworm (*T. molitor*) larvae (184.7 mg/g protein) and pupae (160.8 mg/g protein) [15]. Additionally, the TEAA value for black soldier fly (*Hermetia illucens*) larvae (329.3 mg/g protein [22] is lower than that of Huhu larvae. Among the conventional dietary protein sources, whey (341 mg/g) [23], egg (165 mg/g), beef (329.9 mg/g) [16], and soy (199 mg/g protein) [23] have TEAA values lower than those of HLPE and HPPE. Chickpea (420.4 mg/g protein) [17] has a TEAA value slightly higher than that of HPPE (411.7 mg/g protein) (Table 1). The essential amino acid content of the Huhu grub protein extracts corresponds to the 263 mg/g protein value reported for a good nutritional quality of protein by the UN FAO/WHO [24]. The highest contents of individual essential amino acids for HLPE and HPPE were lysine (76.9, 75.5 mg/g protein), leucine (81.7, 91.3 mg/g protein), isoleucine (50.9, 55.6 mg/g protein), and valine (53.9, 64.5 mg/g protein), respectively, which are reported to be involved in muscle development, signal transduction, and energy supply, respectively [22]. Except for valine, the contents of all the other essential amino acids in HLPE and HPPE were similar (*p* > 0.05). The lysine contents of HLPE and HPPE (76.9 and 75.5 mg/g protein, respectively) are higher than the recommended value of 45 mg/g protein [24], indicating that Huhu grubs can complement foods, such as cereals, which are typically low in lysine content [19].

The histidine contents of HLPE and HPPE (20.8, 19.1 mg/g) are higher than the recommended value of 15 mg/g protein [24]. Histidine is an essential amino acid required for the synthesis of several hormones necessary for metabolic functions. However, it is reported that high levels of histidine, being a precursor of histamine, can trigger allergic reactions upon the consumption of high-histidine-containing materials [25]. Interestingly, the levels of glutamine in Huhu larvae (94.0 mg/g protein) and pupae (87.7 mg/g protein) protein extract are higher than those in *T. molitor*, beef, and chickpea (see Table 1). Glutamine, especially its sodium salts, is the main contributor to the umami taste of food. Cysteine, a sulphur-containing amino acid, is considered non-essential but has been listed by the WHO [14] as a required amino acid for both adults and children. The combined contents of cysteine and methionine in HLPE and HPPE (12.3, 15.2 and 17.5, 17.0 mg/g protein, respectively) meet the proposed requirement of 22 mg/g protein [24]. We found that the asparagine levels were low in HLPE and HPPE (2.5 mg/g protein). Asparagine is not an essential amino acid and can react with reducing sugars in Maillard-type reactions to form potentially toxic acrylamide [26]. Although studies on the risks of a high intake of specific amino acids are lacking, it is well known that a very high intake of protein and amino acids (>2.4 g/kg body weight) can lead to renal glomerular sclerosis and accelerated osteoporosis [26].

### 3.3. Essential Amino Acid Index (EAAI) of Huhu Grub Protein Extracts

The essential amino acid index (EAAI) is a nutritional index for amino acids. It is used to determine the nutritional quality of protein based on all the essential amino acids in comparison to either a reference protein or human requirements. The EAAI for both HLPE and HPPE was determined to be 3.3 and 3.4, respectively (Table 1). The EAAI values of the Huhu larvae and pupae protein extracts were higher than those of *T. molitor* (1.6 for larvae, and 1.3 for pupae) and beef (3.0) but lower than that of chickpea (3.5) (Table 1). Good-quality protein is reported to have an EAAI of above 0.9, a useful protein a value of 0.8, and ab incomplete protein value below 0.7 [27]. HLPE and HPPE can therefore be classed as good-quality proteins. The EAAI is an acceptable index of biological value, specifically when utilised to evaluate a single source of protein, and this is why nutritionists recommend combining different protein sources in diets to achieve a balanced protein intake and optimum EAAI, providing a balanced amino acid complement [27]. Unlike the protein digestibility-corrected amino score (PDCAAS) method, which accounts for the first limiting amino acid, the EAAI method accounts for all essential amino acids, rendering it a comprehensive nutritional index for the evaluation of various food proteins. However, although the EAAI does not account for protein digestibility, it is a good indicator for the evaluation of a protein material prior to conducting either digestibility or feeding trials [27].

### 3.4. Crude Protein, True Protein, and Nitrogen-to-Protein Conversion Factors

Table 2 shows the ‘true protein’ content of the Huhu grub protein extracts based on the sum of amino acid residues, expressed as g/100 g DW, which was 70.1 for HLPE and 75.7 for HPPE. These values are higher than those of cricket (*Acheta domesticus*) (54.9), mealworm (*T. molitor*) (50.9), and locust (*Locusta migratoria*) (46.6), expressed as g/100 g DW [18]. The ‘true protein’ content is considered a better dietary measure of food protein, since this value stems directly from the amino acid composition. On the other hand, crude protein does not distinguish between protein nitrogen and non-protein nitrogen and, therefore, often leads to an overestimation [9]. It is still debated as to whether a partial correction by the subtraction of the non-protein nitrogen content from the total nitrogen content before the calculation of the crude protein (using a nitrogen-to-protein conversion factor of 6.25) provides an accurate analysis [18]. However, the ‘true protein’ values are expected to be lower than the crude protein data [28], and this is what was observed for the Huhu grub protein extracts in the present study, where ‘true protein’ values of 70.1, and 75.7 g/100 g DW were observed and were lower than the crude protein contents determined, being 72.1 and 76.5 g/100 g DW, respectively, using the new the conversion factors, as discussed below.

The *k*_P_ and *k*_A_ values of the Huhu grub protein extracts are shown in Table 2. The *k*_A_ conversion factor is the pure protein conversion factor calculated using the protein nitrogen content (Naa). The *k*_A_ values of *T. molitor*, *A. domesticus*, and *L. migratoria* were 5.4, 5.3, and 5.3, respectively [18], which are lower than the 6.2 and 6.4 values obtained for the Huhu grub extracts. According to Mosse [29], the most accurate way to calculate a nitrogen-to-protein conversion factor is to base it on the amino acid composition. In protein extracts, the non-nitrogenous protein is expected to be low, and measurements of *k*_A_ are, therefore, the most appropriate for such extracts. A similar *k*_A_ value of 6.3 was reported for soy protein extracts [30]. Mathematically, *k*_P_ = *k*_A_ if no non-protein nitrogen compounds are present in a sample, and N_total_ = N_protein_. Hence, it is reported that *k*_A_ is mostly applied when no other nitrogen compounds are present besides protein, as may be the case for protein extracts [18].

The nitrogen-to-protein factor, *k*, can be considered as the average of the total nitrogen present in the protein material, such as that extracted from insects (N_total_), and the total protein nitrogen (N recovered from the amino acid, N_protein_). It is also expected that the lower the levels of non-protein nitrogen are, the higher the *k*_A_ and, to some extent, the *k*_P_ values will be, with the N_total_ usually being higher than the N_protein_. The new conversion factors determined in the present study for the Huhu larvae and pupae protein extracts (6.1 and 6.4, respectively) differ from the 6.25 value commonly used in the literature. These nitrogen-to-protein conversion factors obtained for HLPE (6.1) and HPPE (6.4) were higher than those reported for *A. mellifera* larvae (5.6), *A. mellifera* pupae (4.9), and *S. gregaria* (4.5) [8]. The higher conversion factors are likely due to the fact that the Huhu grub protein extracts were enriched in protein and depleted in non-protein nitrogen compounds.

Moreover, the *k*_A_ values reported in the present study were slightly higher, because glutamine and asparagine were also determined, as well as aspartic acid and glutamic acid. Further, Boisen, et al. [31] reported that correction for amide N is not always necessary for calculating *k*_P_, as the molecular weight of amide forms are similar to that of the acid forms. For example, asparagine and aspartic acid are similar (132.12 and 133.11 g/mol, respectively), and the N_total_ includes the amide N. Using the conversion values newly calculated in the present study provides crude protein contents of 72.1 and 76.5 g/100 g DW, respectively, for Huhu larvae and pupae protein extracts (Table 2). The new *k* values also indicate protein recovery values of 72.1 and 76.5 g/100 g DW for HLPE and HPPE, indicating that the Huhu grub extracts are enriched in protein.

The HLPE and HPPE were prepared and studied to evaluate their suitability, based on the measured parameters, for their potential use as ingredients in future food products. Nitrogen conversion factor calculations based on whole insects and a low *k*_A_ value might result in an overestimation of the protein content, which could be a point for consideration, particularly if the food formulated is designed for infants [30]. A recent report (WHO, 2020) [32] proposed that research reports on protein materials should include a description of the sample clarifying whether it is a crude preparation of natural product material or a protein extract, which would assist the investigator in ascertaining the suitability of using a particular conversion factor (whether *k*_A_, *k*_P_, or *k*).

### 3.5. Characterisation and Techno-Functionalities of Huhu Grub Protein Extracts

#### 3.5.1. Colour Measurement

The colour of an ingredient can impact consumer perceptions and acceptance when used in food products; therefore, it is an important consideration. The colour characteristics of the Huhu larvae and pupae protein extracts are shown in Table 3 and Figure 2. The appearances of the HLPE and HPPE, based on *L** and *b**, were significantly different (*p* < 0.05), except for *a**. The *L**, *a**, and *b** values of HLPE were 38.4 and 4.0 and those of HPPE were 32.8 and 4.3, respectively. The larvae protein extract was yellow-reddish, and the pupae protein extract was a dull yellow-reddish colour. Silkworm (*Bombyx mori*) larvae protein extract (*L**, *a**, and *b** values of 51.7, 5.8, and 12.4) and silkworm (*B. mori*) pupae protein extract (75.21, 2.11, and 24.67, yellow-reddish colour) [33] had colour values higher than those of the Huhu larvae and pupae protein extracts, indicating that the silkworm protein extracts were lighter in appearance. However, *Protaetia brevitarsis* (*L**, *a**, and *b** of 33.2, 4.37, and 6.9) had a darker yellow-reddish hue [2] compared with *Chondracris roseapbrunner* (*L**, *a**, and *b** of 55.8, 5.5, and 18.2), which had a lighter yellow-reddish hue [33]. The browning indices of HLPE (60.0) and HPPE (66.4) were higher than those of *A. mellifera* (31.8) and *S. gregaria* (35.2) [8]. Our visual observation of the colour suggested that colour-producing chemical reactions might have occurred during the processing. Processing parameters such as the alkalinity level, pI precipitation, temperature, and drying can influence the final colour of insect material powder. Enzymatic or non-enzymatic reactions, such as the Maillard process or polyphenol oxidase (PPO), which can be more favourable at higher temperatures and pH levels, are reported to cause browning [6].

#### 3.5.2. Surface Hydrophobicity of Huhu Grub Protein Extracts

The surface hydrophobicity values of the larvae and pupae protein extract powder suspensions were 35.5 and 36.3, respectively (Table 3) and were significantly similar (*p* > 0.05). These surface hydrophobicity readings are lower than those reported for *T. molitor* larvae protein meal (102.5) [34], *S. gregaria* (213.7), *A. mellifera* (237.3), and whey (468.0) [8]. However, the surface hydrophobicity values of *P. succinate* (22.59) and *C. roseapbrunner* (15.92) [33] were lower than those obtained for HLPE and HPPE in the present study. The levels of hydrophobic amino acids, such as isoleucine (50.9 and 55.6 mg/g) and leucine (81.7 and 91.3 mg/g), found in HLPE and HPPE in the present study could account for the observed hydrophobicity. The surface hydrophobicity of proteins can be affected by the extraction process, amino acid composition, and protein conformation and sequence [33]. Although regions of the hydrophobic amino acid sequence are often buried inside the folded structure of proteins, contributing to a reduction in the surface hydrophobicity [34], denatured proteins can have a greater tendency to aggregate due to the exposure of the hydrophobic regions to the aqueous environment.

#### 3.5.3. Particle Size Distribution of the Extracted Protein Extract

The particle size, expressed as the hydrodynamic diameter (d_h_) of proteins in aqueous dispersion, was determined by a light scattering method (Section 2.7), and the data are reported in Table 3. A system is considered polydisperse when the polydispersity index (PDI) is higher than 0.1 [19]. The data obtained for HLPE (0.5) and HPPE (0.6) are comparable to those reported for the alkaline extraction of *S. gregaria* (0.4) and *A. mellifera* (0.2) [8]. Hence, the HLPE and HPPE samples can be considered polydisperse systems, and the PDIs were not significantly different (*p* > 0.05). As mentioned above, the preparation of protein extracts can lead to protein unfolding and the exposure of the hydrophobic groups, which can lead to the formation of intermolecular hydrophobic bonds that cause further protein aggregation and particle size increases [19], increasing the polydispersity.

#### 3.5.4. Structural Characterisation of the Huhu Grub Protein Extracts by FTIR and DSC

The FTIR spectra of the Huhu larvae and pupae protein extracts are shown in Figure 3. Both HLPE and HPPE displayed five distinct FTIR regions. HPPE was found to have higher-intensity peaks for amide I (C=O stretching bonds), amide II (C-H stretching bonds), and amide A (C-N stretching, N-H bending). In contrast, HLPE had higher amide B (C-N stretching, N-H bending) regions. Amide I represents a beta-sheet, alpha-helix, and random coil structure and any reorganisation of the hydrogen bonds [22]. Hence, the FTIR spectra can indicate whether a highly preserved secondary structure is present. In addition, more pronounced absorbance peaks were found in the ranges of ~3000–2960 cm^−^^1^ (representing aromatic rings, alkenes, and alkanes) and 2934–2959 cm^−^^1^ (representing aromatic, aliphatic, and charged amino acids) [22]. The Huhu larvae and pupae protein extracts displayed FTIR peaks equivalent to those reported for *H. illucens* [19], *R. ferrugineus* [6], *B. mellifica*, and *Ascra cordifera* [22]. A split peak was observed for the amide B group representing C-O stretching between 2800 and 3100 cm^−^^1^ for HLPE and HPPE. Queiroz, Regnard, Jessen, Mohammadifar, Sloth, Petersen, Ajalloueian, Brouzes, Fraihi and Fallquist [19] reported a similar split peak for *H. illucens* and attributed this to the asymmetrical stretching of the proteins. Accessible alpha-helix and beta-sheet secondary structures have been reported to increase the number of protein interfacial interactions with non-polar compounds, and the unfolding of the protein structure exposes the secondary structure, enabling molecular interactions with neighbouring biomolecules [35].

Enthalpic peaks were generated when the Huhu grub protein extracts were heated (see Figure 4A,B), with the thermograms containing unfolding (~160 °C) and solid melting (~200 °C) regions, as reported by Queiroz, Regnard, Jessen, Mohammadifar, Sloth, Petersen, Ajalloueian, Brouzes, Fraihi and Fallquist [19] in the case of other proteins. A higher protein denaturation enthalpy (ΔH) was found in HPPE (54.1 J/g) than in HLPE (37.1 J/g) in the present study (Figure 4A,B). These values are lower than those reported for *H. illucens* larvae (219 J/g), *B. mellifica*, and *A. cordifera* (195 J/g) [22]. The denaturation temperatures (ΔT_d_) for HLPE and HPPE were 0.42 °C (peak 1) and 6.74 °C (peak 2), respectively, and 0.89 °C (peak 1) and 18.21 °C (peak 2), respectively. HPPE had a higher denaturation temperature compared to HLPE. A plausible explanation for the higher enthalpy of HPPE compared to HLPE might be that a higher number of hydrogen bonds need to be broken during protein unfolding in HPPE. The FTIR spectra (Figure 3) show that HPPE had more hydrogen bonds than HPLE, as evidenced by the amide A regions (between 3000 and 3500 cm^−^^1^), where N-H stretching vibrations can be observed, which are known to be dependent on the strength of the hydrogen bonds [36]. A higher number of hydrogen bonds has been correlated with higher enthalpies [22]. Moreover, we reported in our previous study that Huhu pupae had a higher fat content (58%) than the larvae (32–50%) [11]. Therefore, there will likely be higher levels of hydrophobic proteins and, hence, hydrophobic bonds in HPPE. Hydrophobic bonds are typically stronger than hydrogen bonds [37], explaining the higher enthalpy value found for HPPE.

### 3.6. Functionalities of Huhu Grub Extracted Proteins

#### 3.6.1. Protein Solubility Profile

Solubility is an important prerequisite for food proteins because it influences the potential applications of specific proteins and is related to the emulsifying and foaming properties [5]. The solubility levels of HLPE and HPPE as a function of pH were similar (Figure 5). These protein extracts showed a relatively higher solubility at the acidic pH 3 and alkaline pH 10. At the pH of the minimum solubility of the extract, the proteins have reduced electrostatic forces of repulsion and maximised protein precipitation, resulting in minimising solubility [38]. In the present study, HLPE and HPPE were found to have a minimum solubility of around pH 5.

Some other insect protein extracts have also been reported to have a pI of 5. For example, protein extracts obtained from silkworm (*B. mori*) and spider (*Nephila edulis*) were found to be between pH 4.37–5.05 and 6.47, respectively [39]. *Rhynchophorus ferrugineus* had a minimum solubility at pH 4.5, and the highest was found at pH 2, 11.5, and 12 [6], and cricket (*A. domesticus)* protein extract exhibited a minimum solubility between pH 3 and 4. For various food proteins, the pI values are reported to range from 3.5 to 6.5 [3]. Hydrophilic and polar amino acids, such as aspartic acid, glutamic acid, and serine, play vital roles in protein solubility and the net charge as a function of pH [40]. For instance, HLPE and HPPE are both high in glutamic acid (94 and 87.7 mg/g protein) and serine (45.1 and 45.3 mg/g protein) contents, respectively, contributing substantially to their solubility (20–23%).

#### 3.6.2. Foaming Capacity and Stability

The foaming capacity and stability of HLPE and HPPE are shown in Figure 6A,B. The Huhu pupae protein extracts (86.0%) produced a higher foaming capacity than the larvae protein extracts (55.3%) at 60 min. The stability of foam is reported to be influenced by the protein structure, concentration, and ionic strength [4]. The foaming capacity values obtained for the Huhu larvae protein extracts in the present study were higher compared to those obtained for *A. mellifera* larvae (43.3–45%) [8], and the Huhu pupae extracts were lower than *S. gregaria* (grasshopper) proteins (90% at 60 min) [8]. The enriched protein content of the protein extracts could explain the enhanced foamability, and in addition, alkaline and sonication treatment could result in protein unfolding or conformational changes, promoting interaction at the air–water interference [41].

A higher foaming stability was observed in HPPE at all time points (0–150 min). Stable foams (75.3% and 73.1% after 120 min) were observed for both HPPE and HLPE, respectively. The foam stability values of HLPE and HPPE (73.1 and 75.3%) were comparable to that of S. *gregaria* (74.1% at 120 min) [8]. Foam formation is governed by the diffusion, reorganisation, and alignment of molecules at the air–water interface. To exhibit good foaming, a protein must be capable of migrating to the air–water interface, unfolding, and rearranging at the interface.

#### 3.6.3. Water- and Oil-Holding Capacity

The water- and oil-holding capacity is expressed as the amount of water or oil (mL) that can be held per gram of the sample [7]. No information is available to date on the water- and oil-holding capacity of protein extracts from Huhu larvae and pupae. As shown in Table 3, the WHC and OHC of proteins from HLPE (9.1 and 7.9 mL/g) were significantly higher (*p* < 0.05) than those of HPPE (7.3 and 5.1 mL/g).

The water-holding capacity provides important information for the use of a protein extract in product development or food technology applications. Due to the high values of WHC, it can contribute to the capacity of a raw high-protein material for use in certain baked products. In particular, hydrophobic residues that interact with hydrocarbon chains in fat molecules can have an impact on the OHC, according to Sathe, et al. [42]. These factors include the protein quantity, protein type, and amino acid composition of proteins. OHC contributes significantly to an improved palatability, taste retention, and tongue feel [43]. The examined insect species may be utilised in the food sector due to their high OHC, as OHC is required in various food applications, such as bakery goods and meat replacements. Insects such as *T. molitor* and *Gryllodes sigillatus* have higher values of WHC and OHC in their protein extracts (3.95 and 2.74 g/g) compared to whole insects (1.29 and 1.71 g/g) [44]. A similar trend of a higher OHC in pupae was observed in silkworm pupae, compared to larvae [45]. The ability to hold water is an important functionality in food technology and can be influenced by factors such as the amino acid profile, protein conformation, hydrophobicity, and protein concentration. OHC may be linked to the particle size and the porous structure of a material that can enable the physical entrapment of oil [7].

#### 3.6.4. Emulsifying Capacity and Stability

The emulsion capacity values of HLPE (82.2%) and HPPE (77.8%) were significantly similar (*p* > 0.05) (Table 3). On the contrary, *A. mellifera* and *S. gregaria* are reported to have a 100% emulsion capacity [8]. The difference in the ability to form an emulsion between insect protein extracts may be due to a higher protein content, improved solubility, protein hydrophobicity, and protein-enriched fractions. High hydrophobicity increases protein adsorption at the oil–water interface, leading to a reduction in interfacial tension and thus improving the facilitation of droplet breakup and emulsification [46].

Appendix A shows the emulsion stability of the Huhu larvae and pupae protein extracts. HLPE and HPPE attained a stable emulsion of 96.8% after 60 min. This might be due to the protein surface charge of the extracts, which, through electrostatic repulsion, prevented the suspended droplets from coalescing [8]. The Huhu larvae and pupae protein extracts were found to have a higher emulsion stability than alkaline extracts of *S. gregaria* and whey (74.8 and 89.8%, respectively) [8]. From a functionality point of view, the hydrophobic amino acid composition has an important role in the emulsifying capacity of the protein. Therefore, among the various amino acids, alanine, valine, glycine, isoleucine, leucine, phenylalanine, proline, and methionine are known as hydrophobic amino acids [2]. A high protein charge is essential for foaming and emulsion formation due to the need for increased electrostatic repulsion between the charged and absorbed protein at an interface [8].

#### 3.6.5. Coagulation of the Protein Concentrate

Table 3 shows the heat coagulation of the Huhu grub protein extracts at the pI and pH 7. The proportion of soluble proteins in HLPE (4.5%) was lower than that in HPPE (7.5%) at pH 5 but was substantially higher in HLPE (17.6%) compared to HPPE (10.5%) at pH 7. *A. mellifera* larvae protein extract had a higher heat coagulation value at pH 5 (18.2%) and pH 7 (30.2%) compared to HLPE and HPPE. The higher heat coagulation of *A. mellifera* correlates with a higher solubility at pH 7 [8]. The lower heat coagulation of the soluble proteins from the HLPE and HPPE fractions is likely due to the lower solubility of the proteins (see Figure 5) and improved protein surface hydrophobicity (Table 3). Higher coagulation properties are desired for a protein required to form firm gels, with less leaching of the non-coagulated proteins in the system [47].

## 4. Conclusions

The present study provides substantial and novel information on the amino acid composition and the chemical and structural characteristics of protein extract obtained from *P. reticularis* (Huhu) larvae and pupae that are endemic to New Zealand. The protein yield (31.9 and 33.5 % DW), essential amino acid contents (386.7 and 411.7 mg/g protein), and essential amino acid indices (3.3 and 3.4), respectively, for HLPE and HPPE show that Huhu larvae and protein extracts are a good source of high-quality, nutritionally sufficient proteins. The determination of the N_total_ and N_protein_ values of the Huhu grub protein extracts enabled the calculation of new nitrogen-to-protein conversion factors (k) of 6.1 and 6.4, respectively, for HLPE and HPPE. The high protein contents of HLPE (72.1%) and HPPE (76.5%) corroborated the observance of five distinct FTIR bands (amides I, II, III, A, and B) that are unique among protein-containing materials. HPPE had higher-intensity peaks corresponding to amide I, amide II, and amide A, while HLPE had higher amide B. These results suggest a difference in the secondary structures of the two protein extracts. For example, a higher enthalpy (54.1 J/g) was found for HPPE compared to HLPE (37.1 J/g), and this correlated with the presence of a higher number of H-bonds in the amide A region of the HPPE FTIR spectra. The water- and oil-holding capacities were higher in HLPE (9.1 and 7.9 mL/g) compared to 7.3 and 5.1 mL/g in HPPE. However, HPPE had a higher foaming capacity (50.7%) than HLPE (41.7%) at 150 min. Although both HLPE and HPPE generated a stable emulsion (96.8%) after 60 min, the emulsion capacity was 82.2% and 77.8%, respectively. This study showed that HLPE and HPPE have potential for incorporation into aqueous food formulations and could also contribute to the improvement of selected properties of various food products, such as bakery products. The data obtained from the present study suggest that HLPE and HPPE preparations exhibit substantial foaming and emulsion properties (Figure 6 and Appendix A, Table 3). In the food industry, foams are used to improve the texture, consistency, and appearance of food products. The emulsion content is highly desirable for the preparation of doughs, salad dressings, ice cream, and infant foods. To our knowledge, this is the first report of the techno-functional characteristics and amino acid compositions of Huhu larvae and pupae protein extracts. These findings provide important information for the potential use of Huhu protein extracts as quality protein ingredients in food technology applications. Although no reports of allergenicity have been documented for Huhu grubs, this is an aspect that should be studied in the future.

## Figures and Tables

**Figure 1 foods-12-00417-f001:**
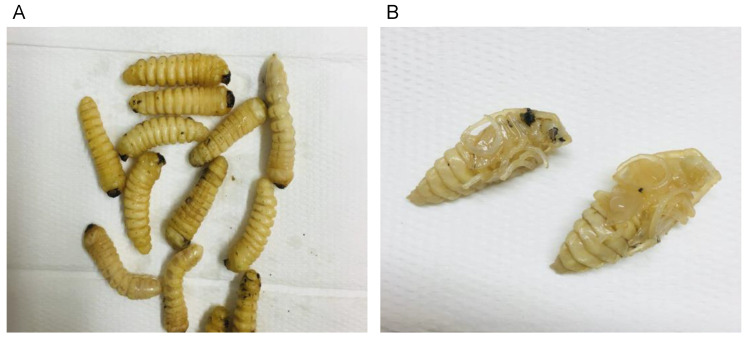
Representative stages of Huhu grub development: (**A**) large larvae and (**B**) pupae.

**Figure 2 foods-12-00417-f002:**
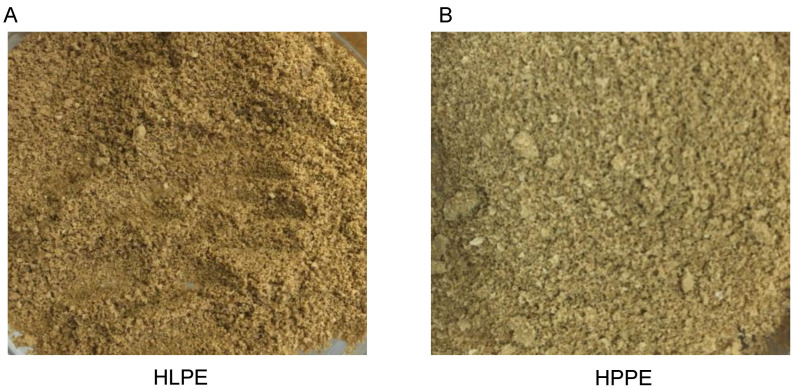
Images of the Huhu larvae (HLPE) (**A**) and pupae (HPPE) (**B**) protein extract powders.

**Figure 3 foods-12-00417-f003:**
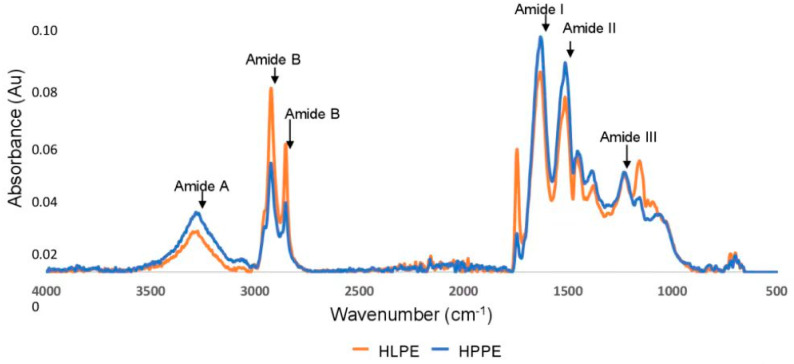
FTIR spectra of the Huhu larvae and pupae protein extracts.

**Figure 4 foods-12-00417-f004:**
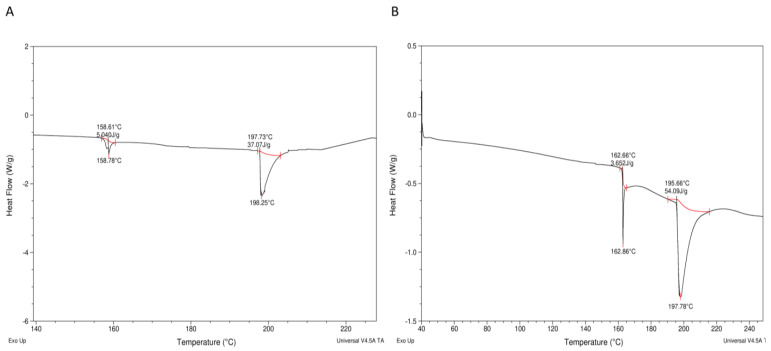
Differential scanning calorimetry (DSC) thermograms for the protein extracts from HLPE (**A**) and HPPE (**B**).

**Figure 5 foods-12-00417-f005:**
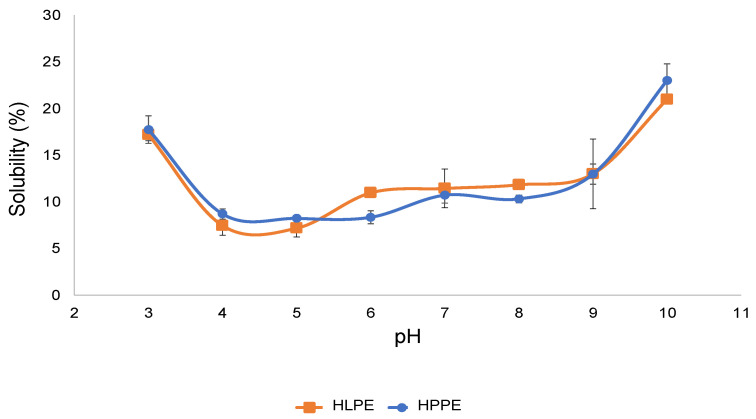
The solubility of the Huhu larvae and pupae protein concentrates at various pH levels.

**Figure 6 foods-12-00417-f006:**
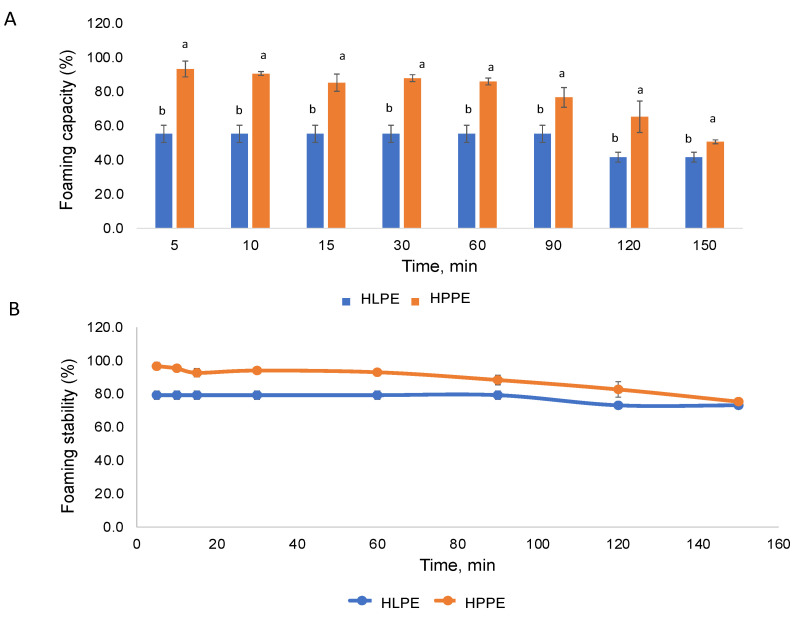
The foaming capacity (**A**), for each time point, means with different superscripts (a,b) are significantly different (*p* < 0.05), as determined by the post hoc Tukey’s honestly significant difference (HSD) test, and foam stability (**B**) of the Huhu larvae and pupae protein extracts at different time points.

**Table 1 foods-12-00417-t001:** Amino acid composition (mg/g protein) of Huhu grub protein extracts and *Tenebrio molitor* larvae and pupae protein extracts in comparison to beef and chickpea protein extracts, with the daily essential amino acid requirements for an adult human.

Amino Acid	Huhu Larvae Protein Extract	Huhu Pupae Protein Extract	*T. molitor* Larvae	*T. molitor* Pupae	Beef	Chickpea	Adult Daily Requirement 1985 FAO/WHO/UN
Essential							
Histidine (His)	20.8 ± 6.4	19.1 ±2.3	22.4	19.0	29.4	27.1	15
Isoleucine (Ile)	50.9 ± 1.8	55.6 ± 3.6	19.7	16.1	38.4	78.9	30
Phenylalanine (Phe)	42.7 ± 1.5	47.7 ± 3.4	18.8	15.3	30.9	60.8	30
Threonine (Thr)	42.2 ± 1.9	40.9 ± 2.5	21.5	20.3	34.3	38.5	23
Lysine (Lys)	76.9 ± 2.4	75.5 ± 4.3	32.3	31.5	66.6	73.7	45
Leucine (Leu)	81.7 ± 1.9	91.3 ± 6.1	29.7	22.1	61.8	78.9	59
Valine (Val)	53.9 ^b^ ± 2.3	64.5 ^a^ ± 4.4	34.4	32.1	44.8	46.8	39
Methionine (Met)	17.5 ± 1.2	17.0 ± 1.1	5.9	4.4	23.7	15.7	22 (Met + Cys)
Total essential amino acid (TEAA)	386.7 ± 9.1	411.7 ± 26.8	184.7	160.8	329.9	420.4	263
EAAI	3.3 ± 0.02	3.4 ± 0.03	1.6	1.3	3.0	3.5	
Non-essential							
Aspartic acid (Asp)	96.2 ± 4.4	102.2 ± 7.6	46.4	44.3	-	59.4	
Cysteine (Cys)	12.3 ± 2.7	15.2 ± 0.9	5.5	4.6	10.1	34.9	
Glycine (Gly)	41.8 ± 1.2	39.4 ± 2.9	26.1	24.2	31.0	41.4	
Arginine (Arg)	60.5 ± 1.6	57.7 ± 3.1	33.2	29.3	47.9	86.3	
Alanine (Ala)	54.9 ^a^ ± 1.3	43.9 ^b^ ± 2.8	46.4	44.3	42.2	43.5	
Serine (Ser)	45.1 ± 1.4	45.3 ± 3.6	23.4	20.9	32.0	52.0	
Tyrosine (Tyr)	58.3 ^b^ ± 2.9	81.8 ^a^ ± 4.6	35.0	30.6	27.1	11.8	
Glutamine (Glu)	94.0 ± 23.2	87.7 ± 7.0	65.7	60.4	46.8	83.3	
Proline (Pro)	50.8 ± 3.4	48.9 ± 4.2	40.4	35.1	30.0	42.9	
Taurine (Tau)	7.1 ± 0.2	6.9 ± 0.3	-	-	-	-	
Asparagine (Asn)	2.5 ± 0.2	2.5 ± 0.2	-	-	-	-	
Sum of total amino acids	910.0 ± 23.4	943.1 ± 62.3	-	-	-	-	

Tau = taurine (not incorporated into protein). Tryptophan was not determined in this study. Data for *Tenebrio molitor* were sourced from Yu et al. (2021) [15]; beef (chuck) was sourced from Wu et al. (2016) [16]; and chickpea was sourced from Rafii et al. (2020) [17]. Adult daily amino acid requirements were sourced from the 1985 FAO/WHO/UN (WHO, 1985) recommendations. EAAI = essential amino acid index. All values are expressed as mean ± standard deviation (SD); n = 3. Means with different superscripts (^a,b^) within each row are significantly different (*p* < 0.05), as determined by the post hoc Tukey’s honestly significant difference (HSD) test.

**Table 2 foods-12-00417-t002:** True protein content (%, DW), total nitrogen (N_total_), nitrogen recovered from amino acids (N_protein_), *k*_A_ (upper limit), *k*_P_ (lower limit), *k* (nitrogen-to-protein conversion factor), and crude protein content of Huhu larvae and pupae protein extracts.

Huhu Grub Extract Source	True Protein (%, DW)	N_total_	N_protein_	*k* _A_	*k* _P_	Nitrogen-to-Protein Conversion Factor, *k* *	Crude Protein Content (%)	
							New *k*	*k*_P_ (6.25)
Larvae	70.1 ^a^ ± 3.6	11.9 ^b^ ± 0.0	11.3 ^a^ ± 0.5	6.2 ^a^ ± 0.1	5.9 ^a^ ± 0.0	6.1 ^a^ ± 0.2	72.1 ^b^ ± 1.7	74.3 ^b^ ± 0.1
Pupae	75.7 ^a^ ± 4.6	12.1 ^a^ ± 0.0	11.8 ^a^ ± 1.5	6.4 ^a^ ± 0.4	6.3 ^a^ ± 0.8	6.4 ^a^ ± 0.4	76.5 ^a^ ± 4.8	75.3 ^a^ ± 0.1

* Calculated as the average of *k*_A_ and *k*_P_. All values are expressed as mean ± standard deviation (SD); n = 3. Means with different superscripts (^a,b^) within each column are significantly different (*p* < 0.05), as determined by the post hoc Tukey’s honestly significant difference (HSD) test.

**Table 3 foods-12-00417-t003:** Techno-functionality and physiochemical characteristics of the Huhu larvae and pupae protein extract powder suspensions.

		HLPE	HPPE
Colour	*L**	38.4 ^a^ ± 1.4	32.8 ^b^ ± 0.6
	*a**	4.0 ^a^ ± 0.4	4.3 ^a^ ± 0.1
	*b**	15.8 ^a^ ± 0.4	14.5 ^b^ ± 0.2
	BI (browning index)	60.0 ^a^ ± 3.3	66.4 ^a^ ± 2.8
Water-holding capacity (WHC) (g/mL)		9.1 ^a^ ± 0.5	7.3 ^b^ ± 0.2
Oil-holding capacity (OHC) (g/mL)		7.9 ^a^ ± 0.8	5.1 ^b^ ± 0.9
Surface hydrophobicity		35.5 ^a^ ± 1.7	36.3 ^a^ ± 1.4
Polydispersity index (PDI)		0.5 ^a^ ± 0.0	0.6 ^a^ ± 0.1
Hydrodynamic diameter (d_h_, nm)		255.6 ^b^ ± 8.1	1078.7 ^a^ ± 22.2
Emulsion capacity (%)		82.2 ^a^ ± 1.9	77.8 ^a^ ± 4.8
Heat coagulation at pH 5 (%)		4.5 ^b^ ± 0.6	7.5 ^a^ ± 0.7
Heat coagulation at pH 7 (%)		17.6 ^a^ ± 3.6	10.5 ^b^ ± 2.3

All values are expressed as mean ± standard deviation (SD); n = 3. Means with different superscripts (^a,b^) within each row are significantly different (*p* < 0.05), as determined by the post hoc Tukey’s honestly significant difference (HSD) test.

## Data Availability

Not applicable.

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
