# Peer review of "Physicochemical Characteristics, Techno-Functionalities, and Amino Acid Profile of *Prionoplus reticularis* (Huhu) Larvae and Pupae Protein Extracts"

_foods, 2023, doi:10.3390/foods12020417_

Round 1
Reviewer 1 Report
1. Line 15: Provide the full names of “FIIR” and “DSC”.
2. Line 16: It should read as “…..larvae (HLPE), and pupae (HPPE) were investigated.”
3. Line 21-22. How about the Td of both protein extracts?
4. Line 22-23. State the protein concentration in the test of foaming property and emulsifying activity.
5. Line 22-23. Both ability and stability of emulsifying and foaming properties are needed.
6. Solubility, coagulation, colour, particle size, surface hydrophobicity, and FTIR results were all missing in the Abstract.
7. Line 37-38. Please include the reference “Chaijan, M., Chumthong, K., Kongchoosi, N., Chinarak, K., Panya, A., Phonsatta, N., ... & Panpipat, W. (2022). Characterisation of pH-shift-produced protein isolates from sago palm weevil (Rhynchophorus ferrugineus) larvae. Journal of Insects as Food and Feed, 8(3), 313-324.”
8. Section 2.1 Sample Collection, please provide the pictures of larvae and pupae of Huhu from your own samples.
9. Line 84. Provide the defatting and drying processes.
10. Line 86. Why alkaline extraction was conducted at 60C? Why not cold refinement process? Please clarify this point since it can affect the nutritional quality and discoloration of the resulting protein. The browning reaction, both enzymatic and Maillard reactions, can be promoted under this condition.
11. Please state the pH of the protein solution during solubilization in alkaline solution.
12. Was the pH of the resulting protein extract re-adjusted to pH 7? If not why? Some functionality is poor at pI.
13. Please italicize L*, a*, and b* throughout the text.
14. Line 163. Why only dried powder was tested? Why not the paste/suspension with added water? In the application of food protein, normally it is not in the dried state.
15. Line 225-237. Why not thermal gelation? Why coagulation?
16. Please recheck all figures due to the low resolution and too small to read.
17. Line 402-403. Regarding the color, please provide the mechanism of brown discoloration of the powder, like PPO activation during extraction and Maillard reaction during extraction and drying or else.
18. How about the lipid and protein oxidations of the protein extracts? Also, the storage stability and consumer acceptability?
19. Line 596. Please provide the example of food technology applications.
20. Please use this article to compare in the Introduction and discussion regarding the nutritional value.
Author Response
We thank the Reviewer for their comments about our manuscript.
Line 15: Provide the full names of “FIIR” and “DSC”.
Done.
- Line 16: It should read as “…..larvae (HLPE), and pupae (HPPE) were investigated.”
Done.
- 3. Line 21-22. How about the Td of both protein extracts?
This has been added to the main text.
- Line 22-23. State the protein concentration in the test of foaming property and emulsifying activity.
Done.
- Line 22-23. Both ability and stability of emulsifying and foaming properties are needed.
Done.
- Solubility, coagulation, colour, particle size, surface hydrophobicity, and FTIR results were all missing in the Abstract.
According to the author guidelines the abstract should be about 200 words maximum. Therefore, we have added representative data in our abstract.
- Line 37-38. Please include the reference “Chaijan, M., Chumthong, K., Kongchoosi, N., Chinarak, K., Panya, A., Phonsatta, N., ... & Panpipat, W. (2022). Characterisation of pH-shift-produced protein isolates from sago palm weevil (Rhynchophorus ferrugineus) larvae. Journal of Insects as Food and Feed, 8(3), 313-324.”
Done.
- Section 2.1 Sample Collection, please provide the pictures of larvae and pupae of Huhu from your own samples.
Done. This has been added.
- Line 84. Provide the defatting and drying processes.
Done.
- Line 86. Why alkaline extraction was conducted at 60C? Why not cold refinement process? Please clarify this point since it can affect the nutritional quality and discoloration of the resulting protein. The browning reaction, both enzymatic and Maillard reactions, can be promoted under this condition.
Thank you for raising this point. In Section 2.15 of our original submitted manuscript we state “……protein was extracted by alkaline extraction and isoelectric pH precipitation, according to a method reported by Mishyna, Martinez, Chen and Benjamin [8], with slight modification.”
We aimed to use the same conditions as used in other publications reported in the literature, to enable comparison of the results- (https://doi.org/10.1016/j.foodres.2018.08.098, https://doi.org/10.1016/j.fbp.2015.06.003, https://doi.org/10.1111/jfpe.13362)
- Please state the pH of the protein solution during solubilization in alkaline solution.
Done. Line 100 has been updated.
- Was the pH of the resulting protein extract re-adjusted to pH 7? If not why? Some functionality is poor at pI.
Yes, all analyses carried out in solution were adjusted to pH 7.
- Please italicize L*, a*, and b* throughout the text.
Done.
- Line 163. Why only dried powder was tested? Why not the paste/suspension with added water? In the application of food protein, normally it is not in the dried state.’
Making a dry powder is considered beneficial as it enhances material preservation, storage life and handling, and can assist with meeting regulatory importation requirements. Protein powder can be applied in a variety of food applications, as outlined briefly in Section 1 and Section 4 of the manuscript. The focus of the present study was to evaluate the powder. The following link is provided in relation to this:
https://www.sciencedirect.com/science/article/pii/B9780124105409000028
- Line 225-237. Why not thermal gelation? Why coagulation?
We are led to believe that both terms have effectively the same meaning. We have opted for the term coagulation, so as to be consistent with the Mishyna reference as that is used for comparison in the current MS-
https://www.sciencedirect.com/science/article/pii/S0963996918307130?via%3Dihub#bb0125
- Please recheck all figures due to the low resolution and too small to read.
This has been checked and updated.
- Line 402-403. Regarding the color, please provide the mechanism of brown discoloration of the powder, like PPO activation during extraction and Maillard reaction during extraction and drying or else.
Thank you for raising this point. The following has been added to the manuscript: “Our visual observation of colour suggested that colour producing chemical reactions might have occurred during processing. Processing parameters such as alkalinity level, pI precipitation, temperature, and drying can influence the final colour of insect material powder. Enzymatic or non-enzymatic reactions, such as the Maillard process or polyphenol oxidase (PPO), which can be more favourable at higher temperatures and pH levels, are reported to cause browning [6].”
- How about the lipid and protein oxidations of the protein extracts? Also, the storage stability and consumer acceptability?
We recognise the importance of the consideration of lipid and protein oxidation, storage stability and consumer acceptability, however, these aspects are outside the scope of our study reported in the current manuscript.
- Line 596. Please provide the example of food technology applications.
The manuscript has been modified accordingly.
- Please use this article to compare in the Introduction and discussion regarding the nutritional value.
The suggested article has been included in discussion where appropriate in the manuscript.
Reviewer 2 Report
The work is a serious start for the inclusion of protein from the larvae and pupae of Prionoplus reticularis in the spectrum of protein sources on a scientific basis. I have just one suggestion – in the description of the material collecting it is stated "...small and large Prionoplus reticularis larvae were collected...". How were the pupae collected? By leaving the larvae to pupate? And how "big and large" in mm were the larvae? Technical note - for the readability of the work, it would be beneficial if all the mentioned insect species were listed by their full Latin name while first mentioning the species (Tenebrio molitor) and abbreviations (T. molitor) should be used only in subsequent occurrences of the species' name in the text. This should be unified for all species throughout the work, and also if citations from other works are used. 374 A recent report [WHO, 202031] – probably should be the “WHO report [31]”.Author Response
The work is a serious start for the inclusion of protein from the larvae and pupae of Prionoplus reticularis in the spectrum of protein sources on a scientific basis.
We thank the Reviewer for their positive comments about our manuscript.
- I have just one suggestion – in the description of the material collecting it is stated "...small and large Prionoplus reticularis larvae were collected...". How were the pupae collected? By leaving the larvae to pupate?
We thank the Reviewer for spotting this. The sentence has ben modified accordingly.
- Prionoplus reticularis large larvae and pupae were collected from dead Pinus radiata (Pine) logs at Flagstaff, Three Mile Hill, which is a locality near Dunedin in the Otago Region (South Island) of New Zealand (latitude 45.8656 and longitude 170.3785).
And how "big and large" in mm were the larvae?
This information has been added to Line 81-82.
- Technical note - for the readability of the work, it would be beneficial if all the mentioned insect species were listed by their full Latin name while first mentioning the species (Tenebrio molitor) and abbreviations ( molitor) should be used only in subsequent occurrences of the species' name in the text. This should be unified for all species throughout the work, and also if citations from other works are used.
We thank the Reviewer for the comment. The use of the Latin names has been reviewed.
- 374 A recent report [WHO, 202031] – probably should be the “WHO report [31]”.
We thank the Reviewer for spotting this. This has been fixed.
Reviewer 3 Report
Dear authors,
I have thoroughly reviewed the whole manuscript; the ms seems to be written nicely covering relevant literature. Here are a few comments, which need to be respond. Once the authors go through the comments and do the needful the ms can be accepted for publication.
i) In section 2.1 Sample collection, “Huhu pupae may be recognised by their flexible abdominal segments….” Authors need to give some authentic data for identifying Prionoplus reticularis pupae and larvae. They may submit samples to zoological survey-related bodies or by a renowned taxonomist.
ii) As the author mainly analysed protein and its characterization, they are also suggested to check allergens from the raw pupae water extract and the extracted protein for food safety issues.
Author Response
Dear authors, I have thoroughly reviewed the whole manuscript; the ms seems to be written nicely covering relevant literature. Here are a few comments, which need to be respond. Once the authors go through the comments and do the needful the ms can be accepted for publication.
We thank the Reviewer for their positive comments about our manuscript.
1. In section 2.1 Sample collection,“Huhu pupae may be recognised by their flexible abdominal segments….” Authors need to give some authentic data for identifying Prionoplus reticularis pupae and larvae. They may submit samples to zoological survey-related bodies or by a renowned taxonomist.
To date, there is only one species known of the genus Prionoplus. Moreover, the features and appearance of the grubs we collected for study were the same as those published in Edwards (1961a and 1961b)*, two pioneering studies on Huhu grubs. Therefore, we did not do verification by an entomologist.
Edwards, J. S. (1961a). Observations on the ecology and behaviour of the huhu beetle, Prionoplus reticularis White.(Col.: Ceramb.). Transactions of the Royal Society of New Zealand, 88, 733-741.
Edwards, J. S. (1961b). Observations on the biology of the immature stages of Prionoplus reticularis White (Col. Ceramb.). Transactions of the Royal Society of New Zealand, 88, 727-731.
In addition, our research group has published 2 papers on the same grub prior to this study, as follows:
Kavle, R. R., Carne, A., Bekhit, A. E. D. A., Kebede, B., & Agyei, D. (2022). Macronutrients and mineral composition of wild harvested Prionoplus reticularis edible insect at various development stages: nutritional and mineral safety implications. International Journal of Food Science & Technology. https://doi.org/10.1111/ijfs.15545
Kavle, R. R., Carne, A., Bekhit, A. E.-D. A., Kebede, B., & Agyei, D. (2022). Proximate composition and lipid nutritional indices of larvae and pupae of the edible Huhu beetle (Prionoplus reticularis) endemic to New Zealand. Journal of Food Composition & Analysis. doi:doi: 10.1016/j.jfca.2022.104578
2. As the author mainly analysed protein and its characterization, they are also suggested to check allergens from the raw pupae water extract and the extracted protein for food safety issues.
We acknowledge relevance of the consideration of allergy. Huhu grubs have been consumed for hundreds of year traditionally by indigenous peoples of New Zealand, and more recently by peoples of many nationalities at ‘wild food’ festivals held regularly in New Zealand. No cases of sensitivity to or allergenicity in relation to consumption of the grubs has been documented. As a result, allergenicity was not considered to be a focus in the present study. "Although no reports of allergenicity have been documented for Huhu grubs, this is an aspect that should be studied in the future."
The last sentence has been added to the conclusion section.
Reviewer 4 Report
This study provides a significant theoretical basis for the commercial production of edible insects in the future. The paper is so well written that I can hardly make suggestions. It would be nice to add a comparative study of protein and amino acid extraction and commercialization techniques between insects and conventional livestock.
Author Response
This study provides a significant theoretical basis for the commercial production of edible insects in the future. The paper is so well written that I can hardly make suggestions.
We thank the Reviewer for their positive comments about our manuscript.
- It would be nice to add a comparative study of protein and amino acid extraction and commercialization techniques between insects and conventional livestock.
The amino acid content of Huhu grub protein has been compared with conventional protein sources and discussed in section 3.3. Some information about potential uses of the Huhu grub material has been added in the Conclusion, section 4.